# Exploring Pediatric Perspectives on Crohn’s Disease: A Qualitative Study of Knowledge, Lived Experience, and Self-Management

**DOI:** 10.3390/healthcare13141710

**Published:** 2025-07-16

**Authors:** Sara Azevedo, Luís Rodrigues, Ana Isabel Lopes

**Affiliations:** 1Gastroenterology Unit, Pediatrics Department, Santa Maria University Hospital—ULS Santa Maria, Academic Medical Centre of Lisbon; 1649-028 Lisbon, Portugal; 2Faculdade de Medicina, Universidade de Lisboa, 1649-004 Lisbon, Portugal

**Keywords:** adolescent, Crohn’s disease, health-related quality of life, inflammatory bowel disease, interview, patient-reported outcomes

## Abstract

**Background:** Pediatric Crohn’s Disease (CD) affects more than physical health, influencing emotional well-being, social integration, and developmental milestones, with an impact on disease management. This study aimed to explore adolescents’ lived experiences with CD and identify factors influencing their motivation for self-management. **Methods:** A descriptive, cross-sectional qualitative study was conducted using a semi-structured, self-administered online questionnaire. Participants (n = 10) were adolescents with CD who had been diagnosed for over three years and were recruited from a tertiary pediatric gastroenterology center. Data included demographics, clinical characteristics, IMPACT-III (HRQOL), and PROMIS short forms. Open-ended responses underwent thematic analysis using the framework developed by Braun and Clarke. **Results:** Participants (80% female, median age 16.2 years, median disease duration 4.6 years) were all in clinical remission (median PCDAI = 2) and with good quality of life (median IMPACT-III = 80.7). Six themes emerged: (1) disease knowledge, (2) emotional responses, (3) coping and adaptation, (4) social support, (5) daily life and school impact, and (6) transition to adult care. Most participants demonstrated strong disease literacy and reported effective coping strategies. Emotional responses to diagnosis ranged from relief (60%) to distress (40%); relapses commonly triggered anxiety and fear. Therapeutic changes and disease monitoring were perceived as beneficial (100%) but with concern. Diagnostic procedures were viewed as burdensome by 70% of respondents. School performance and extracurricular participation were negatively affected in 40% during flares. Concerns regarding the future were reported by 40% of participants, with 30% believing that CD might limit life aspirations. While 60% managed their disease independently, 30% relied on parental support. All acknowledged the need for transition to adult care, though readiness varied. **Conclusions:** This study illustrates the overall impact of disease on pediatric CD patients. It reports significant emotional challenges and difficulties, as well as an impact on daily life, despite good disease knowledge. The findings underscore the importance of psychosocial well-being, ongoing mental health assessment, non-invasive monitoring, and holistic care, emphasizing the patient perspective, in managing pediatric CD.

## 1. Introduction

Pediatric IBD is a complex and challenging disease, frequently described as having a more aggressive phenotype compared to adult IBD [1], thus increasing the risk of poor outcomes. Although it can be diagnosed at a very young age [2,3], the diagnosis typically occurs during adolescence, forcing the patient to adapt to a new reality of disease management, regular exposure to a healthcare team, and multiple treatments, while navigating the regular developmental changes in adolescence [4]. The impact of IBD on patients and their families can be profound, affecting not only their well-being but also their overall health and quality of life (HRQOL) [5,6]. Moreover, they may experience significant and chronic stress related to the unpredictable, painful, and potentially embarrassing symptoms associated with IBD [7] and the ongoing management of the disease, leading to maladaptive functioning, poor self-management, and poor health outcomes.

The family environment and support, the role of the caregiver, the social context, and the relationship with the healthcare team all play a crucial role in the effectiveness of the pediatric patient self-management [8] and overall HRQOL.

Several pediatric studies have documented that pediatric IBD patients and their families experience impaired HRQOL in several physical and psychological domains, and also difficulties in autonomy related to disease activity, requiring specific coping strategies [9,10]. Previous research has also shown that adolescent patients’ knowledge regarding their disease, treatments, and the ability to deal with healthcare systems is insufficient [4,11,12,13,14], and that pediatric patients tend to depend on their parents for disease management and medical decision-making. A recent study using questionnaires to describe the knowledge and self-efficacy skills of 80 IBD teenagers [15] further illustrated suboptimal knowledge regarding medications, smoking, and appointment management.

Knowledge of the disease and its management empowers pediatric patients and promotes better self-management and HRQOL, as it helps patients recognize, promote, and increase their ability to meet their own needs, solve their own problems, and mobilize the necessary resources to take control of their lives [16]. The conceptualizations of self-management generally encompass the daily activities that individuals must perform to manage illness effectively, minimize its impact on their physical health and functioning, and cope with any comorbid or resulting psychological symptoms [8,17]. Interestingly, two essential theories support the concept of self-care and self-management: the Middle Range Theory of Self-Care in Chronic Illness [18] and the Theory of Dyadic Illness Management [19]. According to the Middle Range Theory of Self-Care for Chronic Illness, self-care is an active process that maintains and promotes health in situations of chronic conditions, consisting of behaviors that support the stability of the chronic condition, or the actions patients adopt to manage the condition. It also defines self-care monitoring as the process of monitoring for signs and symptoms of disease, and self-care management as the behaviour performed to manage signs and symptoms when they appear. The Theory of Dyadic Illness Management advocates that the management of a disease is a dyadic phenomenon, in which patients and their caregivers work as an interdependent team and engage in specific behaviour to manage the health condition together.

In IBD, knowledge regarding the most effective strategies for self-management interventions is emerging but remains unclear. In a recent systematic review [20] that aimed to clarify the status and efficacy of self-management interventions for IBD (adult population), the authors concluded that interventions that focus on providing information regarding symptom management and an individual participatory intervention may be effective and support self-management behavior in patients with IBD. In the pediatric setting, a systematic review of self-management skills assessment tools available for children with IBD [21] concluded that there is a lack of adequate tools to assess self-management skills, particularly at younger ages. Finally, a recent study aimed to investigate and describe patient self-care and caregiver contributions to self-care in pediatric IBD [17] highlighted the importance of self-care management in achieving better disease control and a better QOL In this study, it is also suggested that improved educational programs, targeting self-care management strategies, considering the patient and their caregiver perspective, must be included in the treatment plan.

This is particularly relevant to the clinical management of pediatric IBD patients, where parents play an essential role as caregivers and primary reporters of the child’s health status and symptoms, and where the clinical team can easily overlook the pediatric patient’s perspective.

The focus of healthcare in IBD is evolving and becoming more comprehensive [22], including the assessment of physical and psychosocial functioning using patient-reported data and, in the pediatric context, respecting and promoting autonomy [23].

In this descriptive and qualitative study, we aimed to explore and illustrate the perspective of pediatric CD patients, considering their disease conceptualization and experience. The patient’s perspective regarding knowledge of self-management strategies was also explored.

## 2. Materials and Methods

### 2.1. Study Design

This was a descriptive and qualitative, cross-sectional, single-site, anonymous survey study using a self-administered online questionnaire, conducted at a single reference center for pediatric gastroenterology. The participants comprised a subsample of pediatric CD patients, selected from those simultaneously enrolled in a larger prospective study [24]. The principal research was designed to evaluate the clinical utility of the PROMIS system by comparing it with existing assessment tools and included a total of 31 patients. The primary study aimed to assess the clinical applicability of the Patient-Reported Outcomes Measurement Information System (PROMIS) by comparing its measures with standard clinical assessment tools, involving a total of 31 patients. For the present analysis, a convenience sample was drawn from this cohort during routine outpatient visits.

Eligibility criteria for inclusion in this subsample were a confirmed diagnosis of CD for at least three years, age ≥12 years, and sufficient verbal and written comprehension of the Portuguese language. Patients (or their caregivers) who declined to provide informed consent or did not meet the linguistic or age criteria were excluded from participation.

Informed consent was obtained from all participants aged 16 years or older, as well as from caregivers of participants under 16 years of age. In addition, assent was obtained from minors under the age of 16. Following consent, participants received an email with a link to the online questionnaire and were instructed to complete it independently at home. Each participant was permitted to access and submit the survey only once. Responses were automatically anonymized and compiled into a secure database.

The survey distribution took place between April and August 2023, and responses were collected within a two-week window following the initial contact.

### 2.2. Questionnaire Data

The questionnaire, developed by the author, consisted of a 45 open-ended and short answer questions (Table 1) aimed at assessing several items related to the experience of CD and its impact, including the following: Knowledge about the disease; Description of the experience of the disease; Adaptation skills to the disease and to disease worsening; Secondary gains from the disease; School accomplishments, satisfaction and motivation; Integration and social support; Perception of the future; Self-efficacy and transition of care. Patients were instructed to “tell their story of living with CD”.

### 2.3. Other Data

Demographic data and disease-related data (Table 2) were collected from the patients’ medical records and included gender, birth date, school level, and extracurricular activities. The levels of education were classified according to the International Standard Classification of Education [25] (ISCED-2011). ISCED 2011 has nine education levels, from level 0 to level 8. Disease-related data included age at diagnosis, years of disease, disease phenotype (Paris Classification) [26], Pediatric Crohn’s Disease Activity Index (PCDAI) [27] (disease activity score), need for steroid treatment, hospitalization, and/or surgery, and current treatment at enrollment.

To measure HRQOL, we used the Impact III [28], a 35-item self-report, IBD-specific measure of HRQOL, with lower scores indicating poorer HRQOL.

Patient-reported outcomes (PROs) were also assessed using short forms of pediatric PROMIS measures: global health, meaning and purpose, cognitive function, life satisfaction, peer relationships, depression, anxiety, pain interference, and fatigue. PROMIS measures are calibrated using a T-score metric, with the mean of the original calibration population equal to 50 [25].

## 3. Statistical Analysis

Descriptive statistics, including the median and interquartile range, as well as absolute and relative frequencies, were used to analyze patient demographics and disease-related characteristics, and to document patients’ perceptions as reported in the survey answers.

A qualitative analysis of the survey data was performed using thematic analysis, following the six-phase approach outlined by Braun and Clarke [29]. The process involved familiarization with the data, coding, theme development, and refinement. Coding was conducted manually and iteratively, allowing for both semantic and latent themes to emerge. Themes were then reviewed and discussed among the research team to ensure coherence and consistency.

Given the exploratory nature of the study, neither formal data saturation nor structured reflexive journaling was employed; however, the research team engaged in ongoing discussions and critical reflection throughout the analytic process to mitigate interpretive bias and enhance the study’s trustworthiness

## 4. Results

Fifteen patients, fulfilling the inclusion criteria, were invited and agreed to participate, and ten responses were obtained (response rate of 66.6%). Table 2 summarizes the demographic and clinical characteristics of pediatric CD patients who participated in the survey.

The sample consisted of 80% females with an average age of 16.2 years (15; 17.6), with upper secondary education (90%). The median age at diagnosis was 11 years (8.1–13.6 years), and the median disease duration was 4.6 years (3.6–6.4 years). Most patients (40%) had ileocolonic involvement, a non-stricturing, non-penetrating disease phenotype (90%), without perianal disease (80%). Evidence of growth delay was observed in 40% of patients; the mean PCDAI at presentation was 38.8 (±16.9). At the time of diagnosis, 10% of patients were on biological treatment.

In this sample, patients experienced symptoms for a median period of 3.6 months (2.4; 5) until CD diagnosis. At diagnosis, 60% were hospitalized, and one patient (10%) needed surgery due to CD fistulizing disease. Biological treatment is the current treatment in 70% of the patients, and all were in remission, with a median PCDAI score of 2.5 (0; 5). In the six months preceding enrollment, the disease remained stable in all patients. The patients expressed a good HRQOL, as reflected by a median IMPACT III score of 80.7 (79.2; 90.0) and an overall good global, mental, and physical health, as reported by median PROMIS scores (Table 3).

A thematic analysis of interview data from ten adolescents diagnosed with Crohn’s disease (CD) yielded six core themes that capture the multidimensional nature of their lived experiences: (1) Knowledge and Understanding of the Disease, (2) Emotional Responses to Diagnosis and Illness, (3) Coping Strategies and Psychological Adaptation, (4) Social Integration and Support, (5) Impact on Daily Life and School Functioning, and (6) Transition to Adult Healthcare Services. Each theme reflects a distinct yet interconnected aspect of the adolescents’ adaptation to chronic illness. Table 4 summarizes and defines the themes.

### 4.1. Knowledge and Understanding of the Disease

This theme refers to the adolescents’ awareness of their medical condition, the implications of their diagnosis, and their understanding of treatment regimens. Most participants demonstrated a high degree of familiarity with the name and nature of their illness. All ten participants were aware of the diagnosis and reported being able to comprehend the information provided by healthcare professionals and family members. Furthermore, nine participants accurately identified their prescribed medications and understood the rationale behind their therapeutic regimen (Appendix A).

Regarding treatment Experience and Disease Monitoring (Appendix A), monitoring procedures (e.g., MRI, endoscopy) were perceived as distressing by 70% of patients due to their invasive nature and duration. Appointments were a source of discomfort for 40%, citing anxiety about potential negative results and disruption to daily life.


*“…. I feel fine, but I’m always a little afraid of what I might hear or if they’re going to tell me that the tests aren’t good…”*



*“…. When I have to perform exams, I get frustrated. I hate being in the hospital for a long time, and the tests end up forcing me to stay there for a long time. I don’t like the environment I’m subjected to…”*



*“…Sometimes it’s annoying because it gets in the way of day-to-day commitments, but I like to be accompanied so that I have an idea of how I’m doing…”*



*“…. I’m afraid that I’m regressing, and as a rule, the treatment is always “worse” than the previous one. But I usually manage as well as I can…”*



*“…I usually don’t mind too much because I’m already in the mindset and I know it’s best for me…”*


### 4.2. Emotional Responses to Diagnosis and Illness

This theme encompasses the emotional impact of receiving a chronic illness diagnosis and the evolving psychological responses during the disease course. Initial reactions included shock, sadness, fear, and, in some cases, relief at finally having an explanation for their symptoms (Appendix A).


*“I felt anguished, scared, and frustrated.”*



*“I remember being relieved… the doctors had found out what I had.”*



*“It took me a while to realize what it meant… I was only 11.”*


Participants also reported experiencing distress related to disease flares, changes in treatment, and uncertainty about the future (Appendix A). For some, these experiences induced feelings of helplessness or psychological burden.


*“When the disease isn’t under control, I feel unable to do anything… it affects me psychologically.”*



*“I’m afraid when the treatment changes… I feel like I might be regressing.”*


### 4.3. Coping Strategies and Psychological Adaptation

This theme captures the range of strategies employed by adolescents to manage the emotional and practical challenges of living with CD. Several participants described adopting adaptive coping mechanisms, including psychological counseling, cognitive reframing, and reduced activity during symptom exacerbation (Appendix A).


*“I went to a psychologist to strengthen my self-esteem.”*



*“I try to prepare myself mentally… remind myself others have it worse.”*


Conversely, others expressed difficulties in coping with the demands of chronic illness, indicating a need for targeted interventions.


*“I haven’t managed to figure out how to deal with the crises.”*



*“I try to cope as best I can, but it’s not always possible.”*


Half of the participants reported experiencing anxiety or loss of energy during disease flares, with 40% finding it difficult to manage relapses. Some expressed emotional fatigue associated with diagnostic procedures.


*“I hate being in hospital… I feel like just another sick person… drained and very discouraged.”*


While changes in therapy were generally viewed positively, 30% expressed concern or apprehension about therapeutic escalation.

### 4.4. Social Integration and Support

This theme relates to the adolescents’ perceptions of social acceptance and the quality of support received from family, peers, and the healthcare team. All participants reported feeling supported by their families and close friends (Appendix A). Most perceived that they were treated similarly to their peers and did not feel socially excluded, and most respondents maintained active friendships (Appendix A).


*“My friends treat me the same.”*



*“Yes, I feel supported by everyone.”*



*“…I think it’s more the other way around. It’s personal life, school and family life that end up making Crohn’s disease more difficult. There are many stressful events throughout life that end up causing many bouts of diarrhea…”*



*“Despite having the disease… I’ve maintained good relationships and even made friends…”*


Nonetheless, 30% reported feeling different and reluctance to disclose their condition due to concerns about pity or stigma.


*“I don’t usually tell almost anyone… I don’t like pity.”*


Some adolescents identified indirect benefits associated with their condition, including healthier family lifestyle changes and academic accommodations.


*“Healthy eating and exercise became part of my family’s habits.”*



*“The teacher let me retake a test… it benefited my grade.”*



*“…I think it’s more the other way around. It’s personal life, school and family life that end up making Crohn’s disease more difficult. There are many stressful events throughout life that end up causing many bouts of diarrhea…”*


In contrast, several participants reported experiencing no positive outcomes and instead emphasized the emotional and physical burdens imposed by the illness.

### 4.5. Impact on Daily Life and School Functioning

This theme explores how CD affects educational engagement, extracurricular participation, and future aspirations. While most participants (n = 9) reported enjoying school, with 70% self-identifying as good students (Appendix A), three indicated that frequent absences and reduced concentration during flares had a detrimental effect on academic performance.


*“…Nothing has changed, because despite having the disease, I can do everything I did before, I’ve kept up my school performance, it’s even improved, I’ve maintained good relationships and even made friends, and relations with teachers have remained the same…”*



*“…As for school performance... when my illness isn’t under control, it’s difficult to stay focused and study for tests, which ends up making my school performance very difficult…”*


However, 40% acknowledged that relapses negatively impacted their performance.


*“It affected my performance because I missed so many periods.”*



*“When the illness isn’t under control, it’s difficult to concentrate.”*


Despite these challenges, all ten participants reported having future goals, none of which had been altered due to their illness. Social lives remained largely intact, and friendships were maintained both inside and outside of the school environment (Appendix A). However, physical education and other activities were sometimes limited by disease-related fatigue and symptoms (Appendix A).


*“When the disease isn’t stable, I can’t even get out of bed.”*



*“Physical education sometimes causes cramps and fatigue.”*


### 4.6. Transition to Adult Healthcare Services

This theme addresses the adolescents’ perceptions of and readiness for transitioning from pediatric to adult care settings. All participants acknowledged the importance of transitioning to adult care. Although most participants had not given the transition substantial thought, many expressed apprehension and emotional discomfort regarding the prospect of leaving the pediatric care team (Appendix A).


*“I try not to think… I’m very afraid of the future.”*



*“I’m used to the pediatric team. I’m afraid I won’t feel as welcome.”*



*“I haven’t given it much thought yet, but I consider it a step of greater responsibility…”*


A few participants associated the transition with increased autonomy or as a marker of maturity, and fifty percent believed that transition should occur only after achieving financial independence from their family.

Regarding the age of transition, 9/10 (90%) of respondents could not define an ideal age, while 3/10 (30%) considered it desirable to be “later,” “when independent,” or dependent on individual characteristics (personalized transition).


*“The later, the better.”*



*“Maybe when I’m an independent adult.”*



*“I have no idea. But I think it depends on each person”.*


### 4.7. Additional Observations: Self-Efficacy and Health Management

Although not initially identified as a primary theme, self-efficacy emerged as a relevant subtheme in participants’ descriptions of their health management practices (Appendix A). While not part of the original thematic framework, the concept of self-efficacy surfaced across multiple interviews. It was retained as an emergent subtheme due to its relevance to participants’ evolving roles in managing their care.

Seven adolescents reported independently managing aspects of their condition, such as taking medications and monitoring symptoms. However, fewer participants felt equipped to handle logistical responsibilities, such as managing medication refills (n = 5) or communicating with healthcare providers (n = 6). None reported using disease-specific digital tools, though six participants described regularly searching for health information online. These findings highlight variability in adolescents’ readiness for autonomous health management and underscore potential gaps in transitional support.


*“I manage my illness myself.”*



*“Yes, I research online.”*


## 5. Discussion

This study provides a comprehensive exploration of the lived experiences of pediatric patients with CD, focusing on their knowledge of the illness, emotional and psychological adaptation, disease management, and its impact across various life domains. We also examined patients’ perspectives on secondary gains, academic, familial, and social functioning, future expectations, and transition to adult care. Furthermore, we explored their understanding and engagement with self-management strategies. Collectively, these insights offer a deeper understanding of what it means to live with CD from the perspective of adolescents.

Our findings indicate that, despite a relatively long disease duration, stable clinical condition, and a good overall HRQOL, many patients described the diagnostic process as complex and emotionally distressing. Challenges remain in coping with disease exacerbations, which can impact academic performance, participation in leisure activities, and physical education. Although only a minority explicitly reported emotional distress during relapses, this likely reflects the visible “tip of the iceberg” and underscores the importance of proactively assessing psychological risk in this population. Encouragingly, most patients expressed an overall optimistic view of their current experience with CD. These results support the current guidelines, emphasizing the importance of psychosocial well-being in the management of pediatric IBD patients.

Improved quality of life and the absence of disability are increasingly recognized as critical treatment goals [22]. It is recommended that the psychosocial status, mental health, and quality of life of children with IBD—and their families—be regularly assessed, as they are at elevated risk for mental health illness related to their condition [30].

A recent study using a qualitative approach and virtual semi-structured interviews [31] aiming to explore the mental health experiences of 21 adolescents and young adults with IBD found that the patients endorsed the relevance of incorporating mental health discussions into routine care and during the transition to adult care. In this study, the participants also identified several factors that promote and impede the integration of IBD into one’s identity, which were considered essential and could be explored in clinical appointments. The patients considered having IBD and transitioning through adolescence to adulthood promote the co-occurrence of psychosocial stressors throughout this period. These results align with our findings, which identified Emotional Responses to Diagnosis and Illness, Coping Strategies, and Psychological Adaptation as relevant themes.

Previous studies have consistently documented the negative psychosocial and HRQOL impacts of pediatric IBD [9,10,26], as well as the need for adaptive coping strategies in response to disease activity [9,10]. In our study, adolescents demonstrated high engagement in disease management, often reporting the independent handling of treatment-related tasks, despite still being under parental care, as they considered themselves capable of managing most tasks related to their illness.

Another interesting finding emerging from the survey responses is the impact on well-being caused by the clinical assessments of disease and treatment changes. Although most patients understand and recognize the importance of disease monitoring and treatment modifications in achieving treatment goals, they report discomfort and fear induced by several exams, as well as the inconvenience of medical appointments and the need to visit the hospital.

Disease monitoring is mandatory in IBD. The recent Treat-to-Target strategy [22] emphasizes frequent reassessment to achieve long-term treatment goals, particularly mucosal healing, which is associated with sustained remission and improved health outcomes. Achievement of mucosal healing is associated with long-term remission and, consequently, better health-related outcomes, with a positive impact on patients’ lives [32]. However, there is no consensus on when to re-evaluate disease activity after inducing remission. Simultaneously, non-invasive monitoring is becoming increasingly significant for assessing disease activity [32], including small-bowel imaging and fecal calprotectin. Telemonitoring IBD is also advancing as a disease monitoring tool [33], including using of PROMs, home-based tests, and wearables devices, with benefits to patients, including improvement in QOL, a reduction in the number of days lost from school and work, better disease knowledge, and a good cost-effectiveness profile [33]. This is also highlighted by this study, where the detrimental impact of invasive testing became evident.

Our study also highlights the importance of including patients’ perspectives on their health conditions, treatments, and disease management, which is the definition of PROs. The inclusion of PROs is recommended as a good model of patient-centered care [22,30].

The results of this study also suggest that patients are aware of the importance of transitioning care, but they still have limitations in conceptualizing it. This is understandable, as the transition from pediatric to adult care represents a vulnerable period for these patients, who are frequently still adapting to all the disease’s particularities and integrating them into their daily lives. Although the correct age for transition remains to be defined, growing evidence suggests that the age should be individualized and determined by the presence of all the skills and knowledge necessary for proper disease management, thus ensuring better outcomes [4,11,12]. The medical team should adequately prepare the pediatric patient for the transition of care as part of a comprehensive care model. Identifying the concerns and needs of patients [34] and their parents. A recent narrative analysis aimed to understand the problems of patients and caregivers during transition [35], including 16 pediatric IBD patients and 10 parents, concluded that both parents and patients struggle with transition, as they adjust to the illness, parents let go, and the young person “grows up”. This study reinforces the need for psychological interventions that address the well-being of parents in transition programs, highlighting a new field of research.

### Study Limitations

This study has several limitations. First, the data collection relied on a hybrid questionnaire–interview design, utilizing online, open-ended questions. While this approach allowed for broader participation, it may have limited the depth of responses compared to traditional face-to-face interviews.

Second, the sample consisted solely of patients with relatively stable disease and excluded those with recent diagnoses. However, we believe that this cohort—due to their longer disease duration and varied experiences, including past relapses—offered a well-rounded and reflective perspective on living with CD and developing coping strategies over time.

Third, the small sample size and predominance of female adolescents limit the generalizability of our findings. This gender and age imbalance may obscure potential differences in how various subgroups experience and adapt to CD. Also, the gender distribution of our sample (80% female) may have influenced the nature of the narratives, particularly in areas related to fatigue, emotional well-being, and self-perception. A recent narrative review on HRQOL in pediatric IBD [36] highlights gender as a relevant factor, with female patients often reporting lower HRQOL in specific domains such as fatigue and psychosocial functioning. As such, the predominance of female voices in our study may reflect gendered experiences of illness and coping.

Lastly, while newly diagnosed patients and those in active disease phases may offer different insights, they may also struggle to provide the kind of reflective understanding this study sought. Therefore, this investigation should be considered a qualitative narrative aimed at capturing the experience of chronic disease in a specific pediatric cohort.

Despite these limitations, this small qualitative study aimed to contribute to a comprehensive understanding of the impact of a chronic illness on a patient’s life, particularly in pediatric patients, where data assessing patients’ knowledge of IBD are emerging [13,15]. This study contributes valuable insight into the multifaceted experiences of adolescents living with CD.

Future research should aim to include more diverse representative samples of the pediatric IBD population, encompassing different CD and ulcerative colitis phenotypes, and integrate patient insights and perspectives into future PROMIS tools.

## 6. Conclusions

In conclusion, our study further provides a comprehensive and lived illustration of the multifaceted experiences of pediatric patients living with CD. Despite a long disease course and good overall knowledge of the disease, patients report significant emotional challenges and difficulties at diagnosis and during periods of disease flare, which affect various aspects of their lives, including academic performance, leisure activities, and sports. These findings are consistent with recent evidence published elsewhere. While most patients demonstrate strong disease management and adaptive skills, as well as resilience and optimism about their current condition, the psychological distress associated with disease relapse underscores the need for ongoing mental health assessment. Our findings highlight the critical importance of psychosocial well-being as a treatment goal and the incorporation of PROs for patient-centered care. In addition, the study underscores the need for non-invasive disease monitoring methods (including telemonitoring tools), especially in pediatric populations. Overall, our study provides valuable and firsthand insights into the pediatric CD experience, highlighting the need for holistic care approaches that address both medical and psychosocial aspects of patient-centered care.

## Figures and Tables

**Table 1 healthcare-13-01710-t001:** Survey guide questions.

Knowledge of the disease Do you know the name of your illness? Do you know what your illness is? Do you understand what the doctors and your family say about your illness? Do you know how this illness is treated? Do you know the names of the medicines you take? Do you understand why you are being treated?
Description of the illness experience How did you feel when you found out you had CD? What has it been like for you to have CD? Have you done anything to make it easier to cope with having CD?
Coping tools/skills and resilience in managing the disease (relapses/changes in treatment regimens, re-evaluation tests) and stressful life events (family, school, personal events) How do you feel when the disease is not under control? How do you deal with it? How do you feel when you have to change your treatment? How do you deal with it? How do you feel when you have to undergo tests (Analysis, MRI, Endoscopy)? How do you cope with the need of undergo testing? How do you feel when you have to come in for appointments? How do you cope? Do you think having CD makes your personal life, school and family life difficult?
Secondary gains from the disease Do you think having CD has brought you any benefits at home, in your family? And at school? And with your friends?
School, school skills, satisfaction with school, motivation to go to school Do you like school? Do you find it difficult to keep up with the subjects? Do you consider yourself a good student? Do you have plans for your future at school? Do you think having CD has affected your school performance in any way? Or your relationship with your classmates? And with teachers? Have you changed your future plans because of your illness?
Social integration (peer acceptance) Do you have friends at school? And outside of school? Do you feel different because you have CD? Do your friends know you have CD? Do you think they treat you differently because of it?
Social support Do you feel supported by your family? And your friends? Do you think you need more support because you have CD?
Extracurricular and social activities Do you feel restricted in any way by CD at school in your extracurricular and/or other leisure activities? In which activities?
Perception of the future Are you worried about your future? Do you think having CD could limit your future?
Self-efficacy: Ability to perform a task successfully Do you know what to do when you run out of medication? Do you know how to contact your healthcare team? Do you know what to do if you get worse from your illness? Do you know the dates of your appointments and tests? Who takes care of your illness, you or your parents? Do you use an IBD app? Do you research your illness online?
Transition of care What do you think about the need to transition to adult care? In your opinion, is there a better age to transition? Do you think the transition should only happen when you are autonomous from your family (e.g., working)?

**Table 2 healthcare-13-01710-t002:** Demographic and disease-related characteristics of the sample at recruitment.

Gender M/F (%)	2/8 (20/80)
Current age, years, median (IQR)	16.2 (15; 17.6)
Level of education (ISCED) ^a^ n (%)	ISCED 2 (lower secondary education): 1 (10) ISCED (Upper secondary education) 3–5: 9 (90)
Extracurricular activities n (%)	4 (40)
Disease duration, years, median (IQR)	4.6 (3.6; 6.4)
Age at diagnosis, years, median (IQR)	11 (8.1; 13.6)
Time to diagnosis, months, median (IQR)	3.6 (2.4; 5)
Paris, age at diagnosis n (%)	A1a (<10 years) 5 (50) A1b (>10 <17 years) 5 (50)
PCDAI ^b^ median (IQR)	37.5 (30; 45.6)
Paris, location n (%)	L2 (Colonic): 2 (20) L3 (Ileocolonic): 4 (40) L3L4a (Ileocolonic+ Upper discase proximal to Ligament of Treitz): 4 (40)
Paris, phenotype n (%)	B1 (non-stricturing non-penetrating): 9 (90) B3 (Penetrating): 1 (10) Perianal disease 2 (20)
Paris growth n (%)	G0 (No evidence of growth delay): 6 (60) G1 (Growth delay): 4 (40)
Need of hospitalization n (%)	
At diagnosis	6 (60)
Readmission during follow-up (N = 6)	3 (50)
In the prior 6 months ^c^	0
Need of surgery n (%)	
At diagnose	1 (10)
In the prior 6 months ^c^	0
Current treatment n (%)	Immunomodulator 3 (30) Anti-TNF alfa ^d^ treatment 5 (50) Ustekinumab ^e^ 2 (20)
Treatment modifications n patients (%)	
During follow-up	4 (40)
In the prior 6 months ^c^	0
Need of corticosteroids n (%)	0
Poor compliance to treatment n (%)	0
PCDAI ^b^ median (IQR)	2.5 (0; 5)
Fecal Calprotectin μg/g, median (IQR)	247.5 (36.3; 823.8)
IMPACT III ^f^ median (IQR)	81.7 (79.1; 90)

^a:^ ISCED—International Standard Classification of Education: ISCED 0–2: Lower secondary education, ISCED 3–5: Upper secondary education, ISCED > 6: Higher educations; ^b^: PCDAI—Pediatric Crohn’s disease Activity index: scoring < 10 was considered in remission; ^c:^ In the 6 months before recruitment; ^d:^ TNF—Tumoral Necrosis Factor: ^e:^ After failing two previous biological treatments; ^f:^ IMPACT III: 35-item self-administered questionnaire of health-related quality of Life in pediatric IBD, score ranges from 35 (poor) to 175 (best).

**Table 3 healthcare-13-01710-t003:** Patient-reported Outcomes (PROMIS measures) at recruitment.

PROMIS Measure	Median (IQR)
Global Health	43.9 (40.1; 51.1)
Depressive Symptoms	45.2 (43.5; 49.1)
Anxiety	44.5 (34.5; 57.4)
Pain Interference	34 (34; 41.1)
Fatigue	47.3 (33.7; 54.6)
Physical Activity	41.2 (39.8; 50)
Life Satisfaction	44.7 (41; 45.3)
Meaning and Purpose	42.7 (38; 48.5)
Cognitive Function	44.2 (40; 53.5)
Peer Relationships	44.9 (38.6; 56.5)

**Table 4 healthcare-13-01710-t004:** Summary of Themes.

Theme	Definition
**1. Knowledge and Understanding**	Awareness and comprehension of the diagnosis, treatment rationale, and communication dynamics.
**2. Emotional Responses**	Initial and ongoing psychological reactions to illness and its impact.
**3. Coping and Adaptation**	Strategies employed to manage emotional and physical aspects of living with CD.
**4. Social Integration and Support**	Quality of interpersonal relationships and perceived social support.
**5. Impact on Daily Life and School**	Effects on educational performance, routine functioning, and future planning.
**6. Transition to Adult Healthcare Services**	Perceptions and concerns related to shifting from pediatric to adult-oriented care.

## Data Availability

The original contributions presented in this study are included in the article/Appendix A. Further inquiries can be directed to the corresponding author.

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
