# Peer review of "Exploring Pediatric Perspectives on Crohn’s Disease: A Qualitative Study of Knowledge, Lived Experience, and Self-Management"

_healthcare, 2025, doi:10.3390/healthcare13141710_

Round 1

Reviewer 1 Report (Previous Reviewer 1)

Comments and Suggestions for Authors

Peer Review Report for Manuscript "Exploring Pediatric Perspectives on Crohn's Disease: A Qualitative Study of Knowledge, Lived Experience, and Self-Management" (healthcare-3705003)

Dear Authors,

Thank you for the opportunity to review your revised manuscript entitled "Exploring Pediatric Perspectives on Crohn's Disease: A Qualitative Study of Knowledge, Lived Experience, and Self-Management." This version demonstrates significant improvements across multiple domains of the study, most notably in the methodological clarity, the articulation of analytical procedures, and the theoretical grounding of key concepts. The use of Braun and Clarke’s thematic analysis framework represents a meaningful advancement from the previous version, and the incorporation of the Middle Range Theory of Self-Care and the Theory of Dyadic Illness Management offers a solid conceptual foundation for interpreting the findings. The structured presentation of six core themes, each illustrated with verbatim quotes, enhances the interpretive coherence and improves the qualitative depth of the work.

The manuscript is now more robust in terms of analytical rigour and better aligned with current qualitative standards. Furthermore, the discussion is enriched by more critical engagement with contemporary literature on pediatric inflammatory bowel disease, patient-reported outcomes, and the psychosocial implications of chronic illness in adolescence. Nonetheless, there remain a few areas that require further refinement to ensure the manuscript meets the highest standards for publication.

Section-by-Section Observations

Introduction

The introduction now offers a richer and more precise discussion of the theoretical and empirical context of the study. The incorporation of the Middle Range Theory of Self-Care and the Theory of Dyadic Illness Management adds meaningful conceptual grounding and addresses the prior critique concerning the absence of a definition or model for self-care. Furthermore, the authors have appropriately revised the framing of the aim, reducing the emphasis on "motivation for self-care" as a primary outcome, which better reflects the actual focus on lived experience and disease conceptualisation.

Materials and Methods:

The methods section has undergone a substantial revision, with the inclusion of a detailed description of thematic analysis as the analytical approach. The six-step process outlined by Braun and Clarke is appropriately cited and described, and the role of manual coding and thematic development is made explicit. The revised text now clarifies the source and structure of the sample, correctly noting its derivation from a larger PROMIS-based project. While the justification for the convenience sample is sound, the section would still benefit from a brief mention of whether the authors engaged in any process of data saturation or reflective journaling to mitigate interpretive bias. The study remains limited by a lack of detail regarding researcher reflexivity, which, if addressed in even one sentence, would enhance the study’s methodological trustworthiness.

Results

The results are now systematically organised into six themes, each accompanied by clear, well-structured narrative summaries and illustrative quotations. This revision effectively resolves the earlier concern regarding thematic coherence and analytic depth. The quotes are well chosen and appropriately support the interpretive claims. The inclusion of supplementary materials is noted, although the main manuscript could benefit from slightly more precise referencing to those materials (e.g., “Table S3” instead of the more generic “Supplementary Material”). A minor suggestion would be to reframe the additional theme on “Self-Efficacy and Health Management” more clearly within the Results section, either by integrating it into an existing theme such as “Coping” or “Transition to Adult Care” or by treating it as an emergent subtheme with justification.

Discussion

The discussion section is conceptually stronger and more nuanced than in the original submission. The authors avoid previous overgeneralisations and now appropriately qualify their claims, especially when describing patients' engagement and optimism. The discussion draws more effectively on recent literature in qualitative paediatric IBD research, including the Allemang et al. (2024) study. The section on disease monitoring and emotional burden is particularly insightful and grounded in relevant evidence. The limitations section is now more comprehensive and reflective, addressing the constraints introduced by the small, homogenous sample and online data collection format. Nonetheless, the discussion could still benefit from a more explicit acknowledgement of how gender distribution might have shaped the nature of narratives, given that 80% of participants were female.

Conclusion

The conclusions are consistent with the findings and strike an appropriate balance between interpretive insight and caution. The authors rightly underscore the need for psychosocially-informed, patient-centred approaches in managing paediatric CD. They also correctly frame their study as exploratory and narrative in scope, thus avoiding overstatement of generalisability. Suggestions for further research, particularly on digital monitoring tools and transition support, are welcome and relevant.

Ethical and Editorial Requirements

The manuscript now fully complies with MDPI requirements for ethical reporting. The protocol number and committee details have been added and presented in structured format. Editorially, the manuscript is clearer and more polished than in the previous version. However, a few residual language issues and minor stylistic inconsistencies persist (e.g., occasional awkward phrasing, slight punctuation errors, and redundant transitions). While these do not substantially impede comprehension, a final language edit by a native English speaker or professional editor is still advisable to ensure publication-ready quality.

Comments on the Quality of English Language

The overall quality of the English language in the revised manuscript has improved significantly. The text is generally clear and coherent, and key methodological and conceptual elements are now articulated with greater fluency. However, a few minor linguistic issues remain, including occasional awkward phrasing, misplaced modifiers, and typographical inconsistencies. A final proofreading by a native English speaker or professional language editor is recommended to ensure precision, improve readability, and bring the manuscript to a fully publication-ready standard.

Author Response

Response to Reviewer

Dear Authors,

Thank you for the opportunity to review your revised manuscript entitled "Exploring Pediatric Perspectives on Crohn's Disease: A Qualitative Study of Knowledge, Lived Experience, and Self-Management." This version demonstrates significant improvements across multiple domains of the study, most notably in the methodological clarity, the articulation of analytical procedures, and the theoretical grounding of key concepts. The use of Braun and Clarke’s thematic analysis framework represents a meaningful advancement from the previous version, and the incorporation of the Middle Range Theory of Self-Care and the Theory of Dyadic Illness Management offers a solid conceptual foundation for interpreting the findings. The structured presentation of six core themes, each illustrated with verbatim quotes, enhances the interpretive coherence and improves the qualitative depth of the work.

The manuscript is now more robust in terms of analytical rigour and better aligned with current qualitative standards. Furthermore, the discussion is enriched by more critical engagement with contemporary literature on pediatric inflammatory bowel disease, patient-reported outcomes, and the psychosocial implications of chronic illness in adolescence. Nonetheless, there remain a few areas that require further refinement to ensure the manuscript meets the highest standards for publication.

Section-by-Section Observations

Introduction

The introduction now offers a richer and more precise discussion of the theoretical and empirical context of the study. The incorporation of the Middle Range Theory of Self-Care and the Theory of Dyadic Illness Management adds meaningful conceptual grounding and addresses the prior critique concerning the absence of a definition or model for self-care. Furthermore, the authors have appropriately revised the framing of the aim, reducing the emphasis on "motivation for self-care" as a primary outcome, which better reflects the actual focus on lived experience and disease conceptualisation.

Materials and Methods:

The methods section has undergone a substantial revision, with the inclusion of a detailed description of thematic analysis as the analytical approach. The six-step process outlined by Braun and Clarke is appropriately cited and described, and the role of manual coding and thematic development is made explicit. The revised text now clarifies the source and structure of the sample, correctly noting its derivation from a larger PROMIS-based project. While the justification for the convenience sample is sound, the section would still benefit from a brief mention of whether the authors engaged in any process of data saturation or reflective journaling to mitigate interpretive bias. The study remains limited by a lack of detail regarding researcher reflexivity, which, if addressed in even one sentence, would enhance the study’s methodological trustworthiness.

Thank you for your thoughtful feedback. In response, we have added a sentence to the Methods section acknowledging the exploratory nature of the study and describing the informal yet deliberate steps taken—such as team discussions and critical reflection—to address researcher reflexivity and strengthen methodological trustworthiness. In section Statistical analysis the following sentence was added (page 8, lines 179-182): “Given the exploratory nature of the study, neither formal data saturation nor structured reflexive journaling was employed; however, the research team engaged in ongoing discussions and critical reflection throughout the analytic process to mitigate interpretive bias and enhance the study’s trustworthiness”

Results

The results are now systematically organised into six themes, each accompanied by clear, well-structured narrative summaries and illustrative quotations. This revision effectively resolves the earlier concern regarding thematic coherence and analytic depth. The quotes are well chosen and appropriately support the interpretive claims. The inclusion of supplementary materials is noted, although the main manuscript could benefit from slightly more precise referencing to those materials (e.g., “Table S3” instead of the more generic “Supplementary Material”). A minor suggestion would be to reframe the additional theme on “Self-Efficacy and Health Management” more clearly within the Results section, either by integrating it into an existing theme such as “Coping” or “Transition to Adult Care” or by treating it as an emergent subtheme with justification.

Thank you for the helpful suggestion. In response, we have treated “Self-Efficacy and Health Management” as an emergent subtheme within the Results section. While it was not part of the initial coding framework, its relevance to adolescent autonomy. The text was revised and a brief justification has also been added to contextualize the subtheme within the study’s exploratory scope (page 12, lines 336.349) “Although not initially identified as a primary theme, self-efficacy emerged as a relevant subtheme in participants’ descriptions of their health management practices (Table S 10). While not part of the original thematic framework, the concept of self-efficacy surfaced across multiple interviews. It was retained as an emergent subtheme due to its relevance to participants’ evolving roles in managing their care.

 Seven adolescents reported independently managing aspects of their condition, such as taking medications and monitoring symptoms. However, fewer participants felt equipped to handle logistical responsibilities, such as managing medication refills (n = 5) or communicating with healthcare providers (n = 6). None reported using disease-specific digital tools, though six participants described regularly searching for health information online. These findings highlight variability in adolescents’ readiness for autonomous health management and underscore potential gaps in transitional support.

“I manage my illness myself.”

“Yes, I research online.”

Furthermore, the reference to the supplementary material is now more clearly identified as Table SX

Discussion

The discussion section is conceptually stronger and more nuanced than in the original submission. The authors avoid previous overgeneralisations and now appropriately qualify their claims, especially when describing patients' engagement and optimism. The discussion draws more effectively on recent literature in qualitative paediatric IBD research, including the Allemang et al. (2024) study. The section on disease monitoring and emotional burden is particularly insightful and grounded in relevant evidence. The limitations section is now more comprehensive and reflective, addressing the constraints introduced by the small, homogenous sample and online data collection format. Nonetheless, the discussion could still benefit from a more explicit acknowledgement of how gender distribution might have shaped the nature of narratives, given that 80% of participants were female.

The authors appreciate the comment. To better acknowledge this limitation the following sentence was added (page 14, lines 438-222): “ Also, the gender distribution of our sample (80% female) may have influenced the nature of the narratives, particularly in areas related to fatigue, emotional well-being, and self-perception. A recent narrative review on HRQOL in pediatric IBD [36] highlights gender as a relevant factor, with female patients often reporting lower HRQOL in specific domains such as fatigue and psychosocial functioning. As such, the predominance of female voices in our study may reflect gendered experiences of illness and coping.”

Conclusion 

The conclusions are consistent with the findings and strike an appropriate balance between interpretive insight and caution. The authors rightly underscore the need for psychosocially-informed, patient-centred approaches in managing paediatric CD. They also correctly frame their study as exploratory and narrative in scope, thus avoiding overstatement of generalisability. Suggestions for further research, particularly on digital monitoring tools and transition support, are welcome and relevant.

The authors appreciate the comments.

Ethical and Editorial Requirements

The manuscript now fully complies with MDPI requirements for ethical reporting. The protocol number and committee details have been added and presented in structured format. Editorially, the manuscript is clearer and more polished than in the previous version. However, a few residual language issues and minor stylistic inconsistencies persist (e.g., occasional awkward phrasing, slight punctuation errors, and redundant transitions). While these do not substantially impede comprehension, a final language edit by a native English speaker or professional editor is still advisable to ensure publication-ready quality.

The authors agree with the comment and have performed a comprehensive review of the text.

Comments on the Quality of English Language

The overall quality of the English language in the revised manuscript has improved significantly. The text is generally clear and coherent, and key methodological and conceptual elements are now articulated with greater fluency. However, a few minor linguistic issues remain, including occasional awkward phrasing, misplaced modifiers, and typographical inconsistencies. A final proofreading by a native English speaker or professional language editor is recommended to ensure precision, improve readability, and bring the manuscript to a fully publication-ready standard.

The authors agree with the comment and have performed a comprehensive review of the text, using the professional version of the tool “Grammarly”

Reviewer 2 Report (Previous Reviewer 2)

Comments and Suggestions for Authors

Thanks for considering each point raised and you have provided detailed responses to all comments. Revisions have been made throughout the manuscript to address the feedback and you have improved the clarity, depth, and overall quality of the paper. Some further comments are provided.

Comment 1. Please check the spelling in keywords CD, might also include  interviews. 

Comment 2. Check spelling line 100 and 102, check line breaks with words for example line 155,159, 166 but is needed all over.

Comment 3. 1.  Use acronyms to indicate Crohn's disease , now you use both in the text.

Comment 4. In result several citations are italic text and several are not, please check according to referencesystem

Comment 5. In study limitation, there is no quality criteria expressed as you are using a qualitative method, for example transferability, thrustworthiness

Comment 6. Please check spelling in abbreviations Cronh’s Disease (CD),

Author Response

  Response to Reviewer

Thanks for considering each point raised and you have provided detailed responses to all comments. Revisions have been made throughout the manuscript to address the feedback and you have improved the clarity, depth, and overall quality of the paper. Some further comments are provided.

Comment 1. Please check the spelling in keywords CD, might also include  interviews. 

The authors agree with the comment and have performed a comprehensive review of the text, using the professional version of the tool “Grammarly”. The word interview was added to the keywords, as suggested.

Comment 2. Check spelling line 100 and 102, check line breaks with words for example line 155,159, 166 but is needed all over.

The authors agree with the comment and have performed a comprehensive review of the text, using the professional version of the tool “Grammarly”.

Comment 3. 1.  Use acronyms to indicate Crohn's disease , now you use both in the text.

The authors recognized the error. The text was amended, and the changes/corrections are highlighted in blue.

Comment 4. In result several citations are italic text and several are not, please check according to referencesystem

The authors acknowledge the comment. The references are known according to the MDPI specification.

Comment 5. In study limitation, there is no quality criteria expressed as you are using a qualitative method, for example transferability, thrustworthiness

Thank you for your thoughtful feedback. In response, we have added a sentence to the Methods section acknowledging the exploratory nature of the study and describing the informal yet deliberate steps taken—such as team discussions and critical reflection—to address researcher reflexivity and strengthen methodological trustworthiness. In section Statistical analysis the following sentence was added (page 8, lines 179-182): “Given the exploratory nature of the study, neither formal data saturation nor structured reflexive journaling was employed; however, the research team engaged in ongoing discussions and critical reflection throughout the analytic process to mitigate interpretive bias and enhance the study’s trustworthiness”

Comment 6. Please check spelling in abbreviations Cronh’s Disease (CD),

The authors recognized the error. The text was amended, and the changes/corrections are highlighted in blue.

Reviewer 3 Report (Previous Reviewer 3)

Comments and Suggestions for Authors

.

Author Response

The authors read the table carefully. However, no comments were made and therefore, no reply was possible

Reviewer 4 Report (New Reviewer)

Comments and Suggestions for Authors

This manuscript demonstrates that despite a long disease course and good overall knowledge of the disease, patients report significant emotional challenges and difficulties at diagnosis and during periods of disease flare, affecting various aspects of their lives, including academic performance, leisure activities and sports in pediatric patients with CD.

This study provides valuable and lived insights into the pediatric CD experience, highlighting the need for holistic care approaches that address both medical and psychosocial patient-centered features. 

The methodology and results are fine and acceptable.

The discussion is well considered and thought provoking.

We should accept this manuscript.

Author Response

The authors appreciate the comments and  sincerely thank the reviewer for their thoughtful and constructive feedback

This manuscript is a resubmission of an earlier submission. The following is a list of the peer review reports and author responses from that submission.

Round 1

Reviewer 1 Report

Comments and Suggestions for Authors

Dear Authors,

Thank you for the opportunity to review your article entitled "Pediatric Patients' Perspectives on Crohn's Disease: Insights into Disease Experience and Motivation for Self-Care " (healthcare-3657020).

The manuscript presents a descriptive, cross-sectional investigation into the lived experience of paediatric patients living with Crohn’s Disease (CD). Through the use of a semi-structured, self-administered online questionnaire, the study seeks to uncover how adolescents conceptualise their disease, respond emotionally and behaviourally to its challenges, and engage with aspects of self-management. The research is situated within the context of an academic medical centre in Lisbon and includes a cohort of ten patients, each with a disease duration of at least three years and currently in remission.

This work clearly aligns with the aims and scope of Healthcare, as it privileges a patient-centred approach and contributes valuable qualitative insight into the psychosocial and developmental dimensions of chronic illness in childhood. By prioritising the voices of young patients, an often marginalised perspective in clinical literature, it addresses a significant gap in the field. The article has considerable potential to inform clinical practice, particularly in designing holistic care strategies and transition programmes that better support adolescent autonomy, psychological well-being, and self-efficacy.

Nonetheless, while the manuscript is both timely and relevant, it would benefit from considerable revision to enhance its methodological rigour, depth of analysis, and overall clarity. There are several limitations that should be addressed before the manuscript can be considered for publication, most notably with regard to the presentation of qualitative findings and the description of analytical methods.

Key Limitations and Methodological Concerns

Despite these merits, there are notable limitations that warrant careful attention. The most pressing issue concerns the methodological treatment of the qualitative data. While the study purports to explore subjective experience, no formal analytical framework (such as thematic analysis or grounded theory) is employed, and the absence of a rigorous qualitative coding strategy diminishes the interpretative robustness of the findings. The authors state that response patterns were reviewed, but this process lacks transparency and fails to meet the reporting standards commonly associated with qualitative research, such as those outlined in the COREQ checklist.

Furthermore, the sample size, although justifiable in the context of qualitative inquiry, remains small and demographically narrow, particularly with respect to gender balance, as 80% of participants were female. This limitation, while acknowledged, has important implications for the transferability of the results, especially in relation to male adolescents, whose illness narratives and coping strategies may differ significantly.

Another concern pertains to the presentation of results. Although the manuscript includes valuable individual excerpts, these are embedded within a narrative structure that lacks thematic coherence. The findings would be more impactful if organised into discrete analytical categories, allowing the reader to discern the recurring themes across participants and thereby gain a clearer understanding of common patterns and divergences in experience. Moreover, the manuscript frequently describes participants' sentiments (e.g., optimism, concern, resilience) without consistently anchoring these interpretations in verbatim responses, potentially introducing interpretative bias.

The discussion section, although thoughtful and well-referenced, would benefit from deeper engagement with current qualitative literature and a more critical reflection on the limitations of the study design. The authors should also be encouraged to revisit the abstract, which currently compresses complex findings into overly generalised statements, and to adopt a structured format consistent with MDPI standards.

Lastly, certain editorial inconsistencies detract from the manuscript’s professionalism. These include duplicated sections in the Author Contributions, minor grammatical errors, and occasional awkward phrasing. Careful revision by a native English-speaking editor or language professional is strongly recommended to ensure fluency and precision.

Section-by-Section Observations

Title and abstract

The abstract is succinct and broadly informative, yet it lacks a clear structure delineating background, methods, results, and conclusions. Given the structured abstract format required by Healthcare, a revision that aligns with this schema would enhance clarity. In particular, numerical findings should be briefly summarised, and a clearer distinction should be drawn between empirical results and interpretative commentary.

Introduction

The introduction is well developed and adequately cited. It effectively contextualizes pediatric inflammatory bowel disease (IBD) within broader trends in disease burden and psychosocial functioning. However, although the term “self-care” is prominently featured in the manuscript title, it is underdeveloped in the body of the text. No conceptual definition or theoretical framework is provided to situate the notion of “self-care” in the context of pediatric Crohn’s disease. A more robust conceptualization of self-care would significantly strengthen the manuscript. By failing to define or anchor the concept of self-care, the manuscript risks weakening the interpretive power of its findings. This would not only clarify the intended scope of the study, but also enhance its contribution to the scientific debate on pediatric chronic disease management.

(it is recommended to review "BMJ Open Gastroenterol. 2024-08-29;11(1):e001510. doi: 10.1136/bmjgast-2024-001510. PMID: 39209770." ) I also recommend emphasizing what the need for this research is, such as a gap in the international literature or the scarcity of direct patient narratives in pediatric IBD research.

Materials and Methods:

The methods section, while sufficiently detailed in terms of recruitment and ethical procedures, does not meet current standards for qualitative research. The absence of an analytical method or coding strategy significantly undermines the credibility of the interpretative claims made later in the manuscript. It is crucial that the authors describe how they derived meaning from the open-text responses, how data saturation was assessed (if at all), and whether any form of triangulation or researcher reflexivity was employed.

Results

The results are rich in content but presented in a manner that is more descriptive than analytical. Rather than simply reporting patient responses, the authors should aim to distil and label themes, clearly showing how illustrative quotations support each theme. This would also help address the issue of redundancy, as several observations (e.g., fear at diagnosis, dislike of tests) are repeated without sufficient synthesis.

Discussion

The discussion is conceptually sound and includes relevant references. However, the authors should avoid overgeneralizations. For example, statements that patients demonstrated “strong engagement” or “optimism” should be critically examined in light of the small, self-selected sample. The limitations section is honest but somewhat underdeveloped. It would benefit from more in-depth reflection on biases introduced by the online nature of the survey and the exclusion of recently diagnosed patients..

Ethical and Editorial Requirements

Ethical approval is mentioned, but not reported in the structured format recommended by MDPI, please include the protocol number and approval date.

Minor Consideration

Below are some minor considerations.

  • I recommend avoiding repetitions of citations in the same period, consider lines 60-61 the citation (8) is reported twice in the same period and in the following period. It is advisable to rephrase the sentence and use it only once at the end.
  • Consider lines 140, 175-176, 284 the text is crossed out and underlined.

Author Response

We sincerely appreciate the reviewers’ insightful and constructive comments, which significantly strengthened the manuscript. Below, we address each comment in detail, referencing the specific changes made to the manuscript. All revisions are highlighted in color in the updated manuscript for ease of review. We are grateful for the reviewers’ insightful suggestions, which have enhanced the overall quality of the manuscript (manuscript ID: healthcare-3657020) 

Please let us know if further clarifications or adjustments are required.

Reviewer 1

Dear Authors,

Thank you for the opportunity to review your article entitled "Pediatric Patients' Perspectives on Crohn's Disease: Insights into Disease Experience and Motivation for Self-Care " (healthcare-3657020).The manuscript presents a descriptive, cross-sectional investigation into the lived experience of paediatric patients living with Crohn’s Disease (CD). Through the use of a semi-structured, self-administered online questionnaire, the study seeks to uncover how adolescents conceptualise their disease, respond emotionally and behaviourally to its challenges, and engage with aspects of self-management. The research is situated within the context of an academic medical centre in Lisbon and includes a cohort of ten patients, each with a disease duration of at least three years and currently in remission.This work clearly aligns with the aims and scope of Healthcare, as it privileges a patient-centred approach and contributes valuable qualitative insight into the psychosocial and developmental dimensions of chronic illness in childhood. By prioritising the voices of young patients, an often marginalised perspective in clinical literature, it addresses a significant gap in the field. The article has considerable potential to inform clinical practice, particularly in designing holistic care strategies and transition programmes that better support adolescent autonomy, psychological well-being, and self-efficacy.

Nonetheless, while the manuscript is both timely and relevant, it would benefit from considerable revision to enhance its methodological rigour, depth of analysis, and overall clarity. There are several limitations that should be addressed before the manuscript can be considered for publication, most notably with regard to the presentation of qualitative findings and the description of analytical methods.

Key Limitations and Methodological Concerns

1) Despite these merits, there are notable limitations that warrant careful attention. The most pressing issue concerns the methodological treatment of the qualitative data. While the study purports to explore subjective experience, no formal analytical framework (such as thematic analysis or grounded theory) is employed, and the absence of a rigorous qualitative coding strategy diminishes the interpretative robustness of the findings. The authors state that response patterns were reviewed, but this process lacks transparency and fails to meet the reporting standards commonly associated with qualitative research, such as those outlined in the COREQ checklist.

The authors acknowledge and appreciate the comments. The methodology was subjected to a comprehensive revision, which entailed the incorporation of a qualitative analysis (thematic analysis). This analysis yielded the identification of six core themes. A comprehensive review of the results has been conducted, and the findings are presented in the updated version (please refer to Methods and results sections, pages 5-10)

2- Furthermore, the sample size, although justifiable in the context of qualitative inquiry, remains small and demographically narrow, particularly with respect to gender balance, as 80% of participants were female. This limitation, while acknowledged, has important implications for the transferability of the results, especially in relation to male adolescents, whose illness narratives and coping strategies may differ significantly.

We agree that the small sample size is a limitation of this study. This study is part of a doctoral Thesis with global purpose of evaluate the clinical utility and responsiveness of the PROMIS pediatric instruments in pediatric CD patients. to assess the contribution of these instruments in comparison to standard clinical care practices, including the conventional assessment of disease activity and the use of anchor HRQOL questionnaires. The Thesis sought to further reinforce the applicability of PROs in pediatric IBD within clinical settings, where evidence remains emerging, using a validated instrument (PROMIS). In this Thesis, secondary aims were established to complement the main objectives, thereby providing a comprehensive perspective on the use of PROs in pediatric CD, including a qualitative analysis of disease knowledge, including conceptualization and assessment of motivation for self-care behaviors, using a clinical interview, the present study. As only 31 patients were included in the main study, the sample of the present study was a convenience one but also a limited one.

To better clarify the sample the following text was added (MATERIALS AND METHODS Study design, page 4): “The participants comprised a subsample of pediatric Crohn’s disease (CD) patients, selected from those simultaneously enrolled in a larger prospective study (Azevedo et al., 2024). The principal study was designed to evaluate the clinical utility of the PROMIS system by comparing it with existing assessment tools and included a total of 31 patients. The primary study aimed to assess the clinical applicability of the Patient-Reported Outcomes Measurement Information System (PROMIS) by comparing its measures with standard clinical assessment tools and included a total of 31 patients. For the present analysis, a convenience sample was drawn from this cohort during routine outpatient visits.”

Also to better address the sample limitations, a sub-chapter has been dedicated to the limitations of the study. This sub-chapter contains additional information, which can be found on page 12 of the Discussion chapter. : “ Finaly, a recognized limitation of this study is the small sample and the fact that the majority of the group was composed of adolescents and female patients, which makes it difficult to generalize the results to the pediatric population. This is due to the possibility that there are differences in how the different sexes and age groups deal with Crohn's disease”

3- Another concern pertains to the presentation of results. Although the manuscript includes valuable individual excerpts, these are embedded within a narrative structure that lacks thematic coherence. The findings would be more impactful if organised into discrete analytical categories, allowing the reader to discern the recurring themes across participants and thereby gain a clearer understanding of common patterns and divergences in experience. Moreover, the manuscript frequently describes participants' sentiments (e.g., optimism, concern, resilience) without consistently anchoring these interpretations in verbatim responses, potentially introducing interpretative bias.

The authors acknowledge and appreciate the comments. A qualitative analysis approach was used, and the results were organized into six core themes. Please refer to Results chapter (pages 6-10) in the revised manuscript.

4- The discussion section, although thoughtful and well-referenced, would benefit from deeper engagement with current qualitative literature and a more critical reflection on the limitations of the study design. The authors should also be encouraged to revisit the abstract, which currently compresses complex findings into overly generalised statements, and to adopt a structured format consistent with MDPI standards.

The authors appreciate the comments. The abstract was revised and formatted according to instructions to the authors from MDPI standards (please refer to page 1 and 2) The discussion was also revised, overgeneralizations were critically analyzed and removed. The text was improved, and data from qualitative studies was integrated (please refer to the section discussion, pages 11-14). The

5- Lastly, certain editorial inconsistencies detract from the manuscript’s professionalism. These include duplicated sections in the Author Contributions, minor grammatical errors, and occasional awkward phrasing. Careful revision by a native English-speaking editor or language professional is strongly recommended to ensure fluency and precision.

The authors acknowledge the comments. All duplicated sections have been removed. The final version of the manuscript was subjected to a rigorous and comprehensive review process, during which its grammatical aspects were meticulously examined with a view to enhancing clarity and rectifying any identified errors. The revised version of the manuscript now comprises the MDPI standards for research studies, including, Front matter: Title, Author list, Affiliations, Abstract, Keywords. Research manuscript sections: Introduction, Materials and Methods, Results, Discussion, Conclusions; Back matter: Supplementary Materials, Acknowledgments, Author Contributions, Conflicts of Interest and References.

Section-by-Section Observations

Title and abstract

6- The abstract is succinct and broadly informative, yet it lacks a clear structure delineating background, methods, results, and conclusions. Given the structured abstract format required by Healthcare, a revision that aligns with this schema would enhance clarity. In particular, numerical findings should be briefly summarised, and a clearer distinction should be drawn between empirical results and interpretative commentary.

The abstract was totally revised according to the comments and suggestions (Please refer to Abstract section of the revised manuscript)

7- Introduction The introduction is well developed and adequately cited. It effectively contextualizes pediatric inflammatory bowel disease (IBD) within broader trends in disease burden and psychosocial functioning. However, although the term “self-care” is prominently featured in the manuscript title, it is underdeveloped in the body of the text. No conceptual definition or theoretical framework is provided to situate the notion of “self-care” in the context of pediatric Crohn’s disease. A more robust conceptualization of self-care would significantly strengthen the manuscript. By failing to define or anchor the concept of self-care, the manuscript risks weakening the interpretive power of its findings. This would not only clarify the intended scope of the study, but also enhance its contribution to the scientific debate on pediatric chronic disease management.

(it is recommended to review "BMJ Open Gastroenterol. 2024-08-29;11(1):e001510. doi: 10.1136/bmjgast-2024-001510. PMID: 39209770." ) I also recommend emphasizing what the need for this research is, such as a gap in the international literature or the scarcity of direct patient narratives in pediatric IBD research.

The authors acknowledge and appreciate the comment. The title was adjusted. The introduction was revised to include more comprehensive and definition of self-management in IBD. The literature was also revised, and relevant references were added to the body of text, to better contextualize the aim of this study (please refer to the Introduction chapter, pages 3-4). The aim was also revised and the focus on self-management was reduced as it not the main purpose of this study (please refer to page 4)

8- Materials and Methods:The methods section, while sufficiently detailed in terms of recruitment and ethical procedures, does not meet current standards for qualitative research. The absence of an analytical method or coding strategy significantly undermines the credibility of the interpretative claims made later in the manuscript. It is crucial that the authors describe how they derived meaning from the open-text responses, how data saturation was assessed (if at all), and whether any form of triangulation or researcher reflexivity was employed.

The authors acknowledge and appreciate the comments. A qualitative analysis approach was used, and the results were organized into six core themes. Please refer to Results chapter (pages 6-10) in the revised manuscript. “ A qualitative analysis of the data from the survey was preformed using thematic analysis following the six-phase approach outlined by Braun and Clarke (Braun, 2006). The process involved familiarization with the data, coding, theme development, and refinement. Coding was conducted manually and iteratively, allowing for both semantic and latent themes to emerge. Themes were then reviewed and discussed among the research team to ensure coherence and consistency”

9- Results The results are rich in content but presented in a manner that is more descriptive than analytical. Rather than simply reporting patient responses, the authors should aim to distil and label themes, clearly showing how illustrative quotations support each theme. This would also help address the issue of redundancy, as several observations (e.g., fear at diagnosis, dislike of tests) are repeated without sufficient synthesis.

Please refer to response to question 1

10-Discussion The discussion is conceptually sound and includes relevant references. However, the authors should avoid overgeneralizations. For example, statements that patients demonstrated “strong engagement” or “optimism” should be critically examined in light of the small, self-selected sample. The limitations section is honest but somewhat underdeveloped. It would benefit from more in-depth reflection on biases introduced by the online nature of the survey and the exclusion of recently diagnosed patients..

The authors acknowledge and appreciate the comment. The discussion wad revised, overgeneralizations were critically analyzed and removed. The text was improved, and data from qualitative studies was integrated (please refer to the section discussion, pages 11-14) To better address the study limitations, a sub-chapter has been added in the section Discussion. This sub-chapter contains additional information, which can be found on page 12 of the Discussion chapter: “This study has several limitations. First, the data collection relied on a hybrid questionnaire-interview design, utilizing online, open-ended questions. While this approach allowed for broader participation, it may have limited the depth of responses compared to traditional face-to-face interviews.

Second, the sample consisted solely of patients with relatively stable disease and excluded those with recent diagnoses. However, we believe that this cohort—due to their longer disease duration and varied experiences, including past relapses—offered a well-rounded and reflective perspective on living with CD and developing coping strategies over time.

Third, the small sample size and predominance of female adolescents limit the generalizability of our findings. This gender and age imbalance may obscure potential differences in how various subgroups experience and adapt to CD.

Lastly, while newly diagnosed patients and those in active disease phases may offer different insights, they may also struggle to provide the kind of reflective understanding this study sought. Therefore, this investigation should be considered a qualitative narrative aimed at capturing the chronic disease experience in a specific pediatric cohort.”

11- Ethical and Editorial RequirementsEthical approval is mentioned, but not reported in the structured format recommended by MDPI, please include the protocol number and approval date.

The authors acknowledge the comments. The protocol number of the local Ethical Committee was added (please refer to page 19)  “. Prior to the commencement of the study, ethical approval was obtained from the Ethical Committee of Santa Maria University Hospital - CHLN, Academic Medical Centre of Lisbon, Portugal- Internal Approval reference 11/20”

12 Minor Consideration Below are some minor considerations: I recommend avoiding repetitions of citations in the same period, consider lines 60-61 the citation (8) is reported twice in the same period and in the following period. It is advisable to rephrase the sentence and use it only once at the end.Consider lines 140, 175-176, 284 the text is crossed out and underlined.

The authors acknowledge the comments. The repeated citations were removed. All track changes were deleted.

Reviewer 2 Report

Comments and Suggestions for Authors

Thanks for the opportunity to review your paper, some suggestions are provided to enhance your paper

Author Response

Response to Reviewer 2

We sincerely appreciate the reviewers’ insightful and constructive comments, which significantly strengthened the manuscript. Below, we address each comment in detail, referencing the specific changes made to the manuscript. All revisions are highlighted in color in the updated manuscript for ease of review. We are grateful for the reviewers’ insightful suggestions, which have enhanced the overall quality of the manuscript (manuscript ID: healthcare-3657020) 

Please let us know if further clarifications or adjustments are required.

Review- Pediatric Patients' Perspectives on Crohn's Disease: Insights into Disease Experience and Motivation for Self-Care Thanks for reviewing your paper, well structured and interesting as well as important research

1-Title and abstract Suggestion to define method in title:

The authors appreciate the insightful comment. The title was adjusted “ "Exploring Pediatric Perspectives on Crohn's Disease: A Qualitative Study of Knowledge, Lived Experience, and Self-Management"

2- Abstract- be consistent with the objective

The authors appreciate the comments The abstract was revised and formatted according to instructions to the authors from MDPI standards (please refer to page 1 and 2)

3- Introduction

Background/rationale Check the possibility to up-date references, some from 2011, 2012, 2014 etc

The authors appreciate the comments, some of the references were deleted. New references were added as the introduction was also revised (please refer to the chapter Introduction)

4- Objectives This study investigates these patients' conceptualization of the disease and motivations for self-management (abstract) we aimed to explore and illustrate the disease perspective and the motivation for self- management behavior, considering the patients’ disease conceptualization in a selected group of pediatric Crohn’s Disease (CD) patients(in theend of method) we conducted a comprehensive exploration of

the patients' experiences …. (discussion)

Please check for consistency in how the study’s objective is described throughout the text.

The authors appreciate the comments. The aim was also revised and the focus on self-management was reduced as it not the main purpose of this study. The consistency of the objective as also revised throughout the manuscript

5- In a cross-sectional study I would suggest using STROBE guidelines and checklist

Results 10 responses were obtained. You express data in mean and SD, The mean value is sensitive to outliers. If the data is skewed the mean may not accurately represent the "typical" value. Standard deviation assumes symmetry in the data. In skewed distributions, it can give a misleading impression of variability. More appropriate to use Median with Percentiles or Quartiles OR Interquartile Range (IQR) – see Table 3 and Table 4 and in text.

The authors acknowledge the comments. All results are now express in median and IQR. Please refer to the update tables 2 (page 16) and table 3 (page 19). Amends were also preformed throughout the abstract and results sections and are highlighted in color.

6- Discussion

Key results You have Summarised your key results fine Limitations Limitations are expressed in your result, suggest to put with a title Generalisability Are expressed -the female gender predominance, imits the generalizability of the findings.

The authors appreciate the comments To better address the sample limitations, a sub-chapter has been dedicated to the limitations of the study. This sub-chapter contains additional information, which can be found on page 12 of the Discussion chapter. : “ Finaly, a recognized limitation of this study is the small sample and the fact that the majority of the group was composed of adolescents and female patients, which makes it difficult to generalize the results to the pediatric population. This is due to the possibility that there are differences in how the different sexes and age groups deal with Crohn's disease”

7- References Some older references from 2011- 2014, could be updated,

The authors appreciate the comments. Some of the old references, although with relevant information, were deleted

8- Ethical aspects Ethical approval was obtained from the Ethical Committee of Santa Maria

The authors acknowledge the comments. The protocol number of the local Ethical Committee was added (please refer to page 19)  “. Prior to the commencement of the study, ethical approval was obtained from the Ethical Committee of Santa Maria University Hospital - CHLN, Academic Medical Centre of Lisbon, Portugal- Internal Approval reference 11/20”

Reviewer 3 Report

Comments and Suggestions for Authors

Thank you for the opportunity to review this manuscript. The topic is important and the authors clearly aim to highlight the voices of young patients with Crohn’s disease, which is commendable. However, I have several concerns that prevent me from recommending publication.

The main issue is the very small and skewed sample (only 10 participants, mostly female, and with a wide age range). This severely limits the generalizability of the findings. Additionally, many responses are based on open-ended questions without a clear or consistent method of qualitative analysis. This makes interpretation subjective and difficult to assess rigorously.

While the themes explored are important, many of the insights (e.g., impact on school, anxiety around diagnosis, coping strategies) are already well documented in the literature. Given the limited novelty and methodological concerns, I feel the study does not add enough to the field in its current form.

I appreciate the effort behind this work, but I would suggest a more robust design and sample for future studies.

Author Response

Reviewer 3

We sincerely appreciate the reviewers’ insightful and constructive comments, which significantly strengthened the manuscript. Below, we address each comment in detail, referencing the specific changes made to the manuscript. All revisions are highlighted in color in the updated manuscript for ease of review. We are grateful for the reviewers’ insightful suggestions, which have enhanced the overall quality of the manuscript (manuscript ID: healthcare-3657020) 

Please let us know if further clarifications or adjustments are required.

Thank you for the opportunity to review this manuscript. The topic is important and the authors clearly aim to highlight the voices of young patients with Crohn’s disease, which is commendable. However, I have several concerns that prevent me from recommending publication.The main issue is the very small and skewed sample (only 10 participants, mostly female, and with a wide age range). This severely limits the generalizability of the findings. Additionally, many responses are based on open-ended questions without a clear or consistent method of qualitative analysis. This makes interpretation subjective and difficult to assess rigorously .While the themes explored are important, many of the insights (e.g., impact on school, anxiety around diagnosis, coping strategies) are already well documented in the literature. Given the limited novelty and methodological concerns, I feel the study does not add enough to the field in its current form.I appreciate the effort behind this work, but I would suggest a more robust design and sample for future studies.

The authors sincerely appreciate the reviewers’ insightful and constructive comments.  The revised version of the manuscript was subjected to a rigorous and comprehensive review process, during which all sections were update and revised, according to all comments from the reviewers.  We agree that the small sample size is a limitation of this study. This study is part of a doctoral Thesis with global purpose of evaluate the clinical utility and responsiveness of the PROMIS pediatric instruments in pediatric CD patients. to assess the contribution of these instruments in comparison to standard clinical care practices, including the conventional assessment of disease activity and the use of anchor HRQOL questionnaires. The Thesis sought to further reinforce the applicability of PROs in pediatric IBD within clinical settings, where evidence remains emerging, using a validated instrument (PROMIS). In this Thesis, secondary aims were established to complement the main objectives, thereby providing a comprehensive perspective on the use of PROs in pediatric CD, including a qualitative analysis of disease knowledge, including conceptualization and assessment of motivation for self-care behaviors, using a clinical interview, the present study. As only 31 patients were included in the main study, the sample of the present study was a convenience one but also a limited one.

To better clarify the sample the following text was added (MATERIALS AND METHODS Study design, page 4): “The participants comprised a subsample of pediatric Crohn’s disease (CD) patients, selected from those simultaneously enrolled in a larger prospective study (Azevedo et al., 2024). The principal study was designed to evaluate the clinical utility of the PROMIS system by comparing it with existing assessment tools and included a total of 31 patients. The primary study aimed to assess the clinical applicability of the Patient-Reported Outcomes Measurement Information System (PROMIS) by comparing its measures with standard clinical assessment tools and included a total of 31 patients. For the present analysis, a convenience sample was drawn from this cohort during routine outpatient visits.”

A qualitative analysis approach was now used, and the results were organized into six core themes. Please refer to Results chapter (pages 6-10) in the revised manuscript. A qualitative analysis approach was used, and the results were organized into six core themes. The following text was added in the methods section (please refer to statistical analyses, page) :  “ A qualitative analysis of the data from the survey was preformed using thematic analysis following the six-phase approach outlined by Braun and Clarke (Braun, 2006). The process involved familiarization with the data, coding, theme development, and refinement. Coding was conducted manually and iteratively, allowing for both semantic and latent themes to emerge. Themes were then reviewed and discussed among the research team to ensure coherence and consistency”